# Mechanisms of Primary Membranous Nephropathy

**DOI:** 10.3390/biom11040513

**Published:** 2021-03-30

**Authors:** Yan Gu, Hui Xu, Damu Tang

**Affiliations:** 1Department of Surgery, McMaster University, Hamilton, ON L8S 4K1, Canada; guy3@mcmaster.ca; 2Urological Cancer Center for Research and Innovation (UCCRI), St Joseph’s Hospital, Hamilton, ON L8N 4A6, Canada; 3The Research Institute of St Joe’s Hamilton, St Joseph’s Hospital, Hamilton, ON L8N 4A6, Canada; 4The Division of Nephrology, Xiangya Hospital of the Central South University, Changsha 410008, China; xuhuiye@csu.edu.cn

**Keywords:** membranous nephropathy, PLA2R, THSD7A, animal models

## Abstract

Membranous nephropathy (MN) is an autoimmune disease of the kidney glomerulus and one of the leading causes of nephrotic syndrome. The disease exhibits heterogenous outcomes with approximately 30% of cases progressing to end-stage renal disease. The clinical management of MN has steadily advanced owing to the identification of autoantibodies to the phospholipase A2 receptor (PLA2R) in 2009 and thrombospondin domain-containing 7A (THSD7A) in 2014 on the podocyte surface. Approximately 50–80% and 3–5% of primary MN (PMN) cases are associated with either anti-PLA2R or anti-THSD7A antibodies, respectively. The presence of these autoantibodies is used for MN diagnosis; antibody levels correlate with disease severity and possess significant biomarker values in monitoring disease progression and treatment response. Importantly, both autoantibodies are causative to MN. Additionally, evidence is emerging that NELL-1 is associated with 5–10% of PMN cases that are PLA2R- and THSD7A-negative, which moves us one step closer to mapping out the full spectrum of PMN antigens. Recent developments suggest exostosin 1 (EXT1), EXT2, NELL-1, and contactin 1 (CNTN1) are associated with MN. Genetic factors and other mechanisms are in place to regulate these factors and may contribute to MN pathogenesis. This review will discuss recent developments over the past 5 years.

## 1. Introduction

Membranous nephropathy (MN) consists of cases with unknown etiology (primary MN/PMN or idiopathic MN/IMN) and incidences caused by other conditions (secondary MN/SMN) including cancers, infections such as hepatitis B, drug reactions, and autoimmune diseases such as lupus; PMNs and SMNs constitute approximately 75–80% and 20–25% of MN cases, respectively [1,2,3]. MN contributes to approximately 30% of nephrotic syndromes in adults [4,5] with the typical clinical features including proteinuria, hypoalbuminemia, hyperlipidemia, and edema [6,7,8]. The first evidence for MN as a kidney-limited autoimmune disease was derived via the immunization of rats with kidney extracts (Heymann nephritis rats) in 1959 [9]; this animal model was instrumental in the subsequent identification of GP330 or megalin expressed on the podocyte surface as the antigen for membranous glomerulonephritis developed in Heymann nephritis rats [10]. In humans, the podocyte surface antigens associated with IMN were later discovered as the M-type phospholipase A2 receptor (PLA2R) in 2009 [11] and thrombospondin domain-containing 7A (THSD7A) in 2014 [12].

The identification of the anti-PLA2R antibody in patients with primary MN was a major breakthrough in the understanding of MN pathogenesis and the clinical management of these patients. PLA2R, and to a lesser extend THSD7A, are the two major MN antigens expressed on the podocyte surface. Based on studies involving different cohorts, accumulative evidence reveals the presence of anti-PLA2R antibodies (aPLA2R-Ab) and aTHSD7A-Ab in 50–80% and 3–5% of PMN cases, respectively [12,13,14,15,16,17]. Circulating aPLA2R-Ab possesses impressive sensitivity and specificity in MN diagnosis. The level of aPLA2R-Ab correlates with the severity of disease, thus offers prognostic value in the evaluation of treatment response, and as a surrogate marker for monitoring disease remission following therapeutic intervention. Evidence strongly supports the pathological role of aPLA2R-Ab and aTHSD7A-Ab; the passive transfer of aPLA2R-Ab to transgenic mice expressing murine PLA2R specifically in podocytes [18] as well as the transfer of aTHSD7A-Ab to mice lead to the rapid development of MN [19]. The functionality of these auto-antibodies to MN is further supported by the newly emerging treatment of MN via the depletion of B cells using rituximab [20].

In spite of major advancements in knowledge and the clinical management of MN, the disease still exhibits a heterogenous prognosis. For MN patients who require therapeutic intervention, only 60% showed partial or complete remission during a 24 month treatment period with rituximab [21]. Personalized treatment on the initial pathological causes leading to the autoimmunity to PLA2R or THSD7A is not yet feasible, as our understanding on the upstream and downstream events contributing to PLA2R- and THSD7A-associated MN remain limited. This is likely attributed to the current animal models for MN being primarily models of passive antibody transfer; these models do not fully recapitulate the course of disease development in MN patients. Upstream pathological events are likely affected by genetic and other factors that might be associated with sex and aging. MN affects males to females at a 2:1 ratio, with the disease onset at a median age of 4^th^–6^th^ decades [3,22]. In this regard, evidence suggests the involvement of genetic factors and other podocyte antigens in MN pathogenesis [23]. This review will focus on recent developments made in the past five years regarding (1) the mechanisms contributing to PLA2R- and THSD7A-derived autoimmunity in MN pathogenesis and (2) other podocyte antigens that affect MN.

To comprehensively review this topic, we have searched PubMed under the term: “membranous nephropathy” [any field] AND “autoantibody” [any field] AND “Journal article” [publication type] AND “2016–present” [data publication]. A total of *n* = 222 articles were retrieved. After the exclusion of reviews, case reports, and articles not directly relevant to this topic, 60 publications remain and are discussed.

Autoimmune reactions of MN occur at the kidney glomeruli, featuring granular IgG deposition along with the deposition of components in the complement system (the membrane attack complex/MAC of component) in the glomerular basement membrane (GBM) adjacent to podocytes, i.e., the subepithelial region [24]. The depositions are caused by (1) mechanisms leading to the shedding of podocyte antigens to the GBM and (2) the binding of antibodies (Figure 1); the deposition contributes to GBM thickening and damage of the glomerular filtration barrier integrity, leading to proteinuria. The most common podocyte surface antigens are PLA2R and THSD7A; others may also be involved, including cell surface protein contactin 1 (CNTN1), intracellular proteins exostosin 1 (EXT1) and EXT2, as well as the secretory protein NELL-1 (neural epithelial growth factor like-1) (Figure 1). The involvement of these antigens in MN pathogenesis will be discussed in the following sections.

## 2. Contributions of Autoimmunity to PLA2R in MN Pathogenesis

### 2.1. The Association of Anti-PLA2R Antibodies with Primary MN

Following the determination of megalin on the podocyte surface as the antigen for nephrotic syndrome in Heymann nephritis rats [10], efforts over a long period of time resulted in the identification in 2009 of PLA2R as a podocyte antigen to which autoimmunization occurs in human PMN [11]. The importance of this discovery should not be underestimated; it revolutionized the clinical management of MN in terms of diagnosis and therapy assessment, as well as refocusing research effort towards illustrating PLA2R-derived impacts on MN. The prevalence of aPLA2R-Ab in PMN over secondary SMN and its clinical values has been extensively studied [25]. Here, we will provide a brief update on recent progress in the past five years.

Evidence clearly reveals a general association of aPLA2R-Ab with primary MN compared to secondary MN. In a study of a Chinese cohort (*n* = 164) consisting of 84 PMN, 22 SMN, 40 non-MN glomerulonephritis, and 20 healthy individuals, aPLA2R-Ab was detected in 64.6% (53/82) of PMN patients [26], 36.4% (8/22) of SMN cases, and not at all in controls (patients with non-MN glomerulonephritis and healthy individuals) [26]. Similarly, in a population containing 122 PMN (or IMN), 30 SMN, and 100 non-MN nephropathy, 82% of PMN and 16.7% of SMN cases were aPLA2R-Ab+ [27]. Among 252 PMN and 32 SMN cases, aPLA2R-Ab+ was detected in 70.6% (178/252) of PMN patients and 28.1% (9/32) of SMN patients [28]. The detection of aPLA2R-Ab in SMN patients ranging from 16% to 36% indicates a complex association of aPLA2R-Ab with MN, suggesting more intricate mechanisms underlying PLA2R impact on MN pathogenesis.

The presence of aPLA2R-Ab in PMN patients correlates with disease severity. In comparison to patients with aPLA2R-Ab– PMN, those with aPLA2R-Ab+ PMN experienced higher proteinuria and nephritic-range proteinuria (> 3.5/day; *p* < 0.05) [26]. Among the 72 IMN patients with nephrotic syndrome, high level of aPLA2R-Ab correlates with elevations in 24-h total proteinuria prior to and after combinational therapy with prednisone plus cyclosporine A [29]. Serum aPLA2R-Ab correlated better than the glomerular deposition of aPLA2R-Ab with reductions in renal function, including serum albumin, serum creatinine, estimated glomerular filtration rate (eGFR), and proteinuria [30]. In 572 patients with biopsy-proven PMN, patients with serum aPLA2R-Ab (68.5%, 392/572) had higher levels of proteinuria compared to those negative for aPLA2R-Ab [31].

In a cohort of 572 biopsy-confirmed IMN cases, a high level of aPLA2R-Ab was associated with a reduced rate of proteinuria remission following immunosuppressive therapy [31]. In a large cohort of 359 PMN patients, 202 patients were aPLA2R-Ab+ based on ELISA analysis [32]. Among these patients, aPLA2R-Ab level was associated with poor spontaneous remission (odds ratio(OR) 2.2, *p* = 0.011) and poor therapy remission (OR 3.15, *p* = 0.004) [32]. Immune suppressive treatment of PMN patients with cyclophosphamide or tacrolimus (FK506) led to decreases in aPLA2R-Ab along with improvement in renal function evident through elevations in serum albumin [33]. In 285 PMN patients with 12-months follow-up, it was observed that patients positive for aPLA2R and intracellular antigens (aldose reductase, SOD2, and α-enolase) had more than a 4-fold higher risk of reduced renal function defined by eGFR < 60 mL/min per 1.73 m^2^ (OR 4.32, 95% confidence interval (CI) 1.41–13.26, *p* = 0.01) [34]. However, serum aPLA2R is not always associated with poor prognosis. Patients with aPLA2R-Ab+ MN had a 83.9% remission rate based on the reduction of proteinuria to <50% of baseline, compared to 54.5% remission rate for patients with aPLA2R-Ab– MN in response to ≥6-month therapy of either glucocorticoid alone or in combination with immunosuppressant therapy [27]. Collectively, circulating levels of aPLA2R-Ab generally predict treatment response. This notion is supported by several meta-analyses. In 28 studies covering 1235 aPLA2R-Ab+ PMN cases and 407 patients with aPLA2R-Ab– PMN, aPLA2R-Ab level correlated with reductions in renal function and aging [35]. Similar conclusions were also derived in a meta-analysis of 12 reports with 2224 PMN patients; furthermore, aPLA2R-Ab levels were associated with non-remission following immunosuppressive therapy (poor remission rate 2.52, 95% CI 1.79–3.55, *p* < 1 × 10^−5^) [36]. In 2345 PMN patients from 29 studies, patients with aPLA2R-Ab– PMN at biopsy or time of diagnosis had a better chance of disease remission (remission rate (RR) 1.31, 95% CI 1.12-–.46, *p* < 0.05) and for aPLA2R-Ab+ patients, antibody reduction at the completion of immunosuppressive therapy predicted better clinical remission (RR 2.86, 95% CI 1.75–4.69, *p* < 0.05) [37]. The poor remission associated with aPLA2R-Ab was also reported in a total of 11 clinical trials involving 824 PMN patients [38].

Anti-PLA2R antibodies provide diagnostic values to PMN. With a defined cutoff titer of aPLA2R-Ab, the diagnosis of PMN could be achieved with 71% sensitivity and 100% specificity in a population containing 69 IMN cases, 9 SMN cases, 94 patients with non-MN glomerulonephritis, and 286 healthy individuals [39] (Table 1). In a cohort of 57 IMN patients, 62 patients with non-MN glomerulonephritis, and 22 healthy individuals, aPLA2R-Ab levels discriminated IMN patients at a ROC (receiver operating characteristic curve) AUC (area under the curve) value of 0.879; with the optimized cutoff value, diagnosis of PMN, at 82.5% sensitivity and 75% specificity [40] (Table 1). The diagnosis of IMN was reported at 88.1% sensitivity and 96% specificity in a population consisting of biopsy confirmed IMN cases (*n* = 67), 200 patients with other renal diseases, and 36 healthy controls [41] (Table 1). The diagnosis of IMN was reported at 83.9% sensitivity and 99.4% specificity in 155 PMN cases compared to 154 controls [42], as well as at 80.8% sensitivity and 98% specificity in 374 confirmed IMN cases vs. 296 non-MN controls [43] (Table 1).

### 2.2. The Anti-PLA2R Antibody as a Cause of PMN

The robust association of aPLA2R-Ab with PMN, along with the correlation of aPLA2R-Ab levels with PMN severity and treatment response, highlights a possible functional impact of aPLA2R-Ab in PMN pathogenesis. This concept is supported by the rituximab-derived depletion of B cells in treating PMN [21]. The temporal relationship between aPLA2R-Ab and the clinical manifestation of PMN fits well with a causal impact of aPLA2R-Ab on PMN. An association between pre-implant aPLA2R-Ab with recurrent MN following kidney transplantation was observed. In a study of 63 transplantations, patients with aPLA2R-Ab had a higher risk of recurrent MN (rMN, hazard ratio (HR) 1.87, 95% CI 1.16–3.0, *p* = 0.01) [44]. Similarly, pre-implant aPLA2R-Ab predicted rMN following kidney transplantation at 85% sensitivity and 92% specificity [45]. In 33 MN patients without nephrotic proteinuria at the time of diagnosis and treated with blockers of the renin-angiotensin system, it was observed that those with aPLA2R-Ab were at increased risk of developing nephrotic proteinuria (HR 3.66, 95% CI 1.39–9.64, *p* = 0.009) [46]. These recurrent MN cases following kidney transplantation in the presence of aPLA2R-Ab support a causative role of the antibody in PMN pathogenesis. Consistent with these observations, in a recent study of 134 PMN patients from the Department of Defense Repository with longitudinal serum samples available, 44% (59/134) of PMN cases were aPLA2R-Ab+ and the appearance of this antibody could be months to years before MN diagnosis and documented non-nephrotic range proteinuria [47].

Even with accumulated evidence supporting a causative role of aPLA2R-Ab in MN, direct demonstration of this concept via the passive transfer of aPLA2R-Ab into mice has been challenging; as mice do not express endogenous PLA2R on podocytes and the construction of transgenic mice expressing human PLA2R in the podocyte has been difficult [48]. However, the situation has changed lately. Transgenic mice expressing full-length murine PLA2R specifically in the podocytes have been constructed [18]. The passive transfer of rabbit anti-mouse PLA2R antibodies induced a rapid onset of MN in the transgenic mice evident by the development of proteinuria, hypercholesterolemia, and morphological features of MN [18]. Collectively, evidence reveals a causative action of aPLA2R-Ab in MN pathogenesis.

The causative link of aPLA2R-Ab to MN provides a solid basis for targeting IgG4 aPLA2R-Ab in MN therapy. In this effort, a Phase II multi-center clinical trial PRISM (peptide GAM immunoadsorption therapy in primary membrane nephropathy) has been conducted [49]. PRISM aims to remove IgG from 12 patients with biopsy-proven PMN; these patients had nephrotic range proteinuria and an aPLA2R-Ab titer > 170 µ/mL. Safety, tolerance to immunoadsorption therapy, as well as reductions of aPLA2R-Ab along with the improvement of renal functions have been examined. The clinical trial was completed at the end of 2019 (Clinical Trial on Autoimmune Membranous Nephropathy: Immunoadsorption—Clinical Trials Registry—ICH GCP), with the outcome yet to be reported. Nonetheless, a positive outcome is to be expected based on the pathological cause of not only aPLA2R-Ab but also aTHSD7A-Ab in MN. Considering the existence of other MN-causing antigens, this IgG immunoadsorption approach might be an attractive alternative therapy.

## 3. THSD7A as a Podocyte Antigen of Membranous Nephropathy

Antibodies to thrombospondin type-1 domain-containing 7A (THSD7A) were initially reported in 15 out of 154 (9.6%) patients with idiopathic MN in 2014 [12]. The presence of serum aTHSD7A-Ab correlates well with the tissue staining of THSD7A in MN biopsies [50]. In comparison to 50–80% of PMN patients having aPLA2R-Ab, aTHSD7A-Ab contributes to approximately 5% of PMN cases [51,52]. Circulating aTHSD7A-Ab was observed in 1.6% of 192 IMN patients [53], 2% of 578 patients with PMN [54], and 3% in 3 studies containing 258 [55], 1012 [52], or 1276 PMN cases [51]. In a meta-analysis of 10 studies conducted up to the end of 2017 for a total of 4121 PMN patients, the 3% prevalence rate of renal stained THSD7A and circulating aTHSD7A-Ab was increased to 10% among aPLA2R-Ab negative cases [56], consistent with PMN cases being positive for either and rarely for both (PLA2R and THSD7A) antigens [23,57]. In two separate studies involving 1270 (258 + 1012) PMN patients, approximately 1% of cases were positive for both PLA2R and THSD7A [52,55]. The distribution of THSD7A-associated PMN cases does not differ among different ethnic groups [56]. Patients with high serum titers of aTHSD7A-Ab were associated with poor prognosis and did not respond to treatment [52]. The diagnostic efficiency of serum aTHSD7A-Ab towards PMN is associated with low sensitivity but high specificity. In a recent meta-analysis of 10 studies involving 4545 PMN patients, aTHSD7A-Ab had achieved diagnosis efficiency in aPLA2R-Ab negative cases at 8% summary sensitivity and 100% summary specificity [58]. The low sensitivity is likely attributable to the low prevalence of serum aTHSD7A-Ab even in patients with PLA2R-negative PMN. Nonetheless, the high specificity can assist in the non-invasive diagnosis of PMN. Collectively, although THSD7A is not a common MN antigen compared to PLA2R, aTHSD7A-Ab levels correlate with poor clinical performance in patients with THSD7A-associated PMN.

Evidence supports aTHSD7A-Ab as a pathological cause of MN. The immunoadsorption of two patients with aTHSD7A-Ab+ MN, one patient with melanoma, and another with bladder cancer, led to reductions of circulating aTHSD7A-Ab and the improvement of renal function [59]. Furthermore, recurrent MN rapidly developed in kidney transplant patients with THSD7A-associated PMN and enhanced THSD7A staining was detected in the kidney allograft [19]. Passive transfer of this patient’s aTHSD7A-Ab or anti-human THSD7A produced from rabbit led to a rapid onset MN pathology in mice without complement activation [19,60].

## 4. Mechanisms Underlying PLA2R- and THSD7A-Contributed MN Pathogenesis

Compared to the rapid expansion of PLA2R and THSD7A as MN antigens, knowledge on their related mechanisms remains limited. Other factors or pathways are critical for both antigens to induce MN. Gene profiling of mouse podocytes revealed pathways regulating cytoskeleton, cell differentiation, endosomal transport, and peroxisome functions [61]. Disruption of these events in podocytes can result in MN. For instance, the binding of aTHSD7A-Ab to cell expressing THSD7A affects the cytoskeleton [19]. Considering that PMN is a kidney-limited autoimmune disease, alterations in the global immune-, inflammation-, and MGN (membranous glomerulonephritis)-associated triplet (IIMATs) networks were recently identified, which include chemokine signaling, the Jak-STAT pathway, B cell and T cell signaling pathways, and others [62], highlighting the importance of abnormalities in immune processes to MN pathogenesis.

### 4.1. Genetic Factors

Genetic factors are clearly involved in MN. For instance, while the passive transfer of rabbit anti-human THSD7A to BALB/c mice led to proteinuria at day 2, with a dramatic increase in severity up to day 14 compared to mice transferred with preimmune IgG, this did not occur in C57BL/6 mice [60]. The passive transfer of rabbit anti-mouse PLA2R IgG to transgenic mice expressing murine PLA2R in the podocyte induced a rapid onset of MN in the BALB/c strain [18]; it will be interesting to examine whether C57BL/6 mice display similar resistance to PLA2R-induced MN.

The risk loci associated with PLA2R MN are involved in peptide presentation in the immune system. The single-nucleotide polymorphism (SNP) rs2187668 of HLA-DQA1 (human leukocyte antigen class II DQ alpha 1) and rs4664308 of PLA2R are risk alleles of PMN [63]. In a study of 1112 IMN patients and 1020 healthy controls, patients with HLA-DQA1 rs2187668 and PLA2R rs4664308 risk alleles had an 11.13-fold higher PMN risk (*p* = 6.03 × 10^−21^) compared to individuals with the protective genotype at either gene [64]. Impressively, among 26 individuals with both risk alleles, 19 (73%) had serum aPLA2R-Ab compared to none in the 19 individuals with the protective genotype at both alleles [64]. In an Indian cohort of 114 PMN patients, rs2187668 of HLA-DQA1 was a risk factor of serum aPLA2R-Ab [65]. The risk allele of PLA2R rs4664308 in PMN development was also reported by others [66]. Both PLA2R rs4664308 and HLA-DQA1 rs2187668 SNPs are intronic [64]; their impact might be on gene expression. Another pair of intronic risk alleles were rs9272729 of HLA-DQA1 and rs17830558 of PLA2R; individuals with homozygous HLA-DQA1 rs9272729 and heterozygous PLA2R rs17830558 had an 80-fold (OR 79.4, *p* = 7.1 × 10^-5^) higher risk of developing MN compared to individuals without the risk alleles at both genes [67].

Risk locus was also found in HLA-DRB1. The DRB1*1501 and DRB1*0301 allele are independent risk factors of IMN with respective ORs of 4.65 (*p* < 0.001) and 3.96 (*p* < 0.001) [68]. Both risk alleles are associated with higher levels of aPLA2R-Ab [68,69]. Interactions of DRB1*1501 and DRB1*0301 independently with PLA2R rs4664308 were detected; it was suggested that arginine 13 and alanine 71 encoded by DRB1*1501, as well as lysine 71 encoded by DRB1*0301, facilitate the presentation of PLA2R epitopes and thereby enhance aPLA2R-Ab production [68]. The risk allele DRB1*1501 was also identified in a separate study involving PLA2R-associated MN (*n* = 343), non PLA2R-associated MN (*n* = 50), and healthy individuals (*n* = 385) [70]. Additionally, it was recently reported that two twin sisters having the risk alleles of DRB1*1501, DRB1*0301, and DQB1*0602 developed PLA2R-associated IMN [71], consistent with the involvement of the risk alleles of DRB1*1501 and DRB1*0301 in developing PLA2R-associated IMN.

In support of the concept for facilitating PLA2R epitope presentation in aPLA2R-Ab production, increases in plasma cells and regulatory B cells (B_REG_) occurred in PLA2R-associated MN patients; memory B cells contributed to aPLA2R-Ab production [72]. Accumulative evidence supports that risk alleles at HLA-DQA1, HLA-DRB1, and PLA2R rs4664308 are risk factors of PLA2R-associated PMN. Risk alleles for THSD7A-associated MNs remain unclear. Collectively, the involvement of HLA-DQA1 and HLA-DRB1 in PLA2R-associated MN pathogenesis is consistent with the current knowledge for the importance of T-cell dependent and B-cell-mediated autoimmunity to anti-receptor-caused autoimmune diseases [73], which include the receptor PLA2R in PMN.

### 4.2. Epitopes of PLA2R and THSD7A

Both PLA2R (180-kDa) and THSD7A (250-kDa) have a large extracellular N-terminal domain on podocytes, which contain several motifs (Figure 2). The N-terminal fragment of PLA2R consists of a cysteine-rich (CysR) or Richin B domain, a fibronectin-like domain (FnII), and 8 C-type lectin-like domain (CTLD); the THSD7A extracellular fragment contains 21 thrombospondin type 1 (TSP-1) domains and a basic region (Figure 2) [57].

Epitope regions of PLA2R, reacted with patient-derived aPLA2R-Abs, were initially mapped to CysR [74], CTLD1, and CTLD7 [75] (Figure 2A). The 31-mer amino acid peptide of CysR binds to aPLA2R-Abs with high affinity and displays 85% inhibition of the interaction of aPLA2R-Abs with PLA2R [74]. All PLA2R-associated PMN patients have circulating aPLA2R-Abs that recognize this epitope [76]. Patients with aPLA2R-Abs recognizing CTLD1 and/or CTLD7 had more severe nephrotic range proteinuria, were less likely to have spontaneous remission, and showed a higher risk of disease progression to end-stage renal disease (ESKD) compared to those with aPLA2R-Abs to CysR epitope [75]. It was suggested that the CysR epitope is the dominant epitope targeted by anti-PLA2R antibodies, which then extends to CTLD1 and/or CTLD7 epitopes as a result of epitope spreading (Figure 2A) following disease progression [75]. In support of this model, patients with epitope spreading had higher aPLA2R-Abs titer sand reduced rates of remission [77]. Epitope spreading is also associated with treatment outcomes; rituximab treatment of patients with nephrotic syndrome and base-line epitope spreading reduced epitope spreading, aPLA2R-Ab titers, and resulted in remission; on the other hand, nonresponse patients showed persistent epitope spreading [77]. However, the association of epitope spreading with the severity and prognosis of PLA2R-associated PMN appears to be more complex. Anti-PLA2R antibodies to the epitope within the CysR-FnII-CTLD1 was previously mapped [78]. An epitope to CTLD8 was recently observed in 16% of 150 patients with PLA2R MN [79] (Figure 2B). All 150 newly diagnosed patients have aPLA2R-Abs reacting to both the N-terminal region (CysR-FnII-CTLD1) and C-terminal region (CTLD7-CTLD8). With 54 months follow-up for therapy with ACE inhibitors or angiotensin receptor blockers, 89% of patients (133 out of 155) had remission of proteinuria independent of antibodies with domain-specific profiles, i.e., a domain-specific antibody does not predict disease outcome independently of total aPLA2R-Abs [79]. Collectively, whether epitope-unique antibodies will specifically impact PLA2R-assocaited MN needs further investigation.

Epitope utilization on THSD7A remain less clear. Among 31 cases of THSD7A-associated MN, 9 epitope regions across 21 TSP-1 domains were identified, with the N-terminal region d1-d2 (domain 1-domain 2) being most frequently recognized [80] (Figure 2C). This is similar to PLA2R, with the CysR epitope having the most common reaction with aPLA2R-Abs [76]. Serum containing antibodies recognizing more than two epitopes has a higher titer of aTHSD7A-Ab [80], which shares similarities with the epitope spreading model of PLA2R [75]. Intriguingly, the common domain epitope between the 31-mer CysR epitope of PLA2R and a 28-mer sequence within d1-d2 regions (48-–192) of THSD7A was recently reported [81]. While cross activities of autoantibodies to either PLA2R or THSD7A could be detected on the common epitope at peptide level, these cross activities did not occur at the protein level [81]. Nonetheless, the involvement of this common epitope remains a possibility in PMN patients that are positive for both PLA2R and THSD7A autoantibodies. It will be interesting to examine this scenario. Collectively, while epitope knowledge is critical in developing domain-specific immunotherapies to PMN, there is still a long road ahead before it can be applied in clinical settings.

### 4.3. Complement Activation in PLA2R- and THSD7A-Contributed MN

Complements (C3 and C5b-9) are present in immune deposits in PMN [82,83,84]; complement activation is a pathological contributor to MN [85,86]. This concept is supported by the essential role of complement activation in developing proteinuria in Heymann nephritis rats [87,88,89]. Additionally, immunization with human recombinant non-collagenous domain 1 (rh-α3NC1) led to the development of proteinuria and the deposition of C3 and C5b-9 in wild type mice but not in mice with AP deficiency [90]. In patients with PLA2R-associated PMN, the accumulation of C3 and C5b-9 in immune deposits occurs [91,92]. Both aPLA2R-Ab and aTHSD7A-Ab are predominant IgG4s [11,12], a subtype IgG that does not fix component C1q and thus is unable to initiate complement activation through the classical pathway (CP) [89]. However, the CP can contribute to PLA2R-associated MN. For instance, in a case of recurrent MN developed 13 days following kidney transplantation, both the native and graft biopsies displayed depositions of monoclonal aPLA2R-Ab IgG3-κ, C1q, C3, and C5b-9 but not MBL (mannose-binding lectin), indicating CP-mediated complement activation [93]. IgG4 was suggested to bind and activate the lectin pathway (LP) [86]. IgG4 purified from aPLA2R-Ab-positive PMN patients was found to bind MBL and induce cytoskeleton alteration in human podocytes in vitro [94]; in a recent study of complement activation products present in the circulation and urine in 134 biopsy-confirmed PMN patients containing 91 patients with PLA2R-associated PMN, complement activation via LP in the presence of aPLA2R-Ab was suggested [95]. Complement activation through the alternative pathway (AP) is supported by the genetic evidence for the deposition of aPLA2R-Ab IgG4, C3, and C5b-9 in patients with MBL deficiency [96]; additionally, in patients with PLA2R-associated PMN, AP can be utilized via the production of antibodies targeting complement factor H (CFH) [97], a cell surface AP inhibitor [98,99]. Collectively, evidence supports the role of complement activation in PLA2R-associated MN.

Evidence is not clear for a role of complement activation in THSD7A-contributed MN. Passive transfer of aTHSD7A-ab derived from patients to mice induces proteinuria, which does not require C3 deposition, although C3 deposition occurs in response to the deposition of mouse-derived anti-human IgG later [19]. Additionally, the injection of rabbit anti-HSD7A resulted in MN without C3 deposition [60].

### 4.4. Physiological Impact of PLA2R and THSD7A on MN Pathogenesis

The passive transfer of anti-THSD7A IgG to BALB/c and C57BL/6 mice induced IgG deposition in both mouse strains but proteinuria only in the BALB/c mice [19], suggesting different downstream events following immune complex deposition occurring in different mouse strains. THSD7A plays a role in podocyte cell skeleton regulation; the binding of THSD7A IgG derived from patients induced substantial cell cytoskeleton reorganization in mouse primary glomerular epithelial cells and in human embryonic kidney 293 (HEK293) cells ectopically expressing human THSD7A [19]. In human podocytes, THSD7A enhances adhesion, facilitates attachment to collagen type IV-coated surfaces, and reduces migration ability [100], supporting its role in regulating the podocyte skeleton. THSD7A is expressed at the basal surface of podocytes in humans and mice [19,101] and in the foot processes proximate to slit diaphragms [102]. The binding of aTHSD7A-Ab can damage the integrity of the filtration barrier, resulting in proteinuria.

The functions of PLA2R remains unclear. In a recently established podocyte-specific mPLA2R (murine PLA2R) mouse model, mPLA2R was detected at the slit diaphragm of podocyte foot processes [18], which implies that the binding of the anti-PLA2R antibody may cause podocyte injury; this may in part contribute to proteinuria. PLA2R was reported to be associated with the annexin A2-S100A10 complex at the podocyte surface and Ca^2+^ enhances this association [103]. Nonetheless, whether this association is sensitive to aPLA2R-Ab and its contributions to MN pathogenesis warrants further investigation.

It remains unclear how certain factors or pathways are involved in leading to the accumulation of soluble PLA2R and THSD7A into the subepithelial region and the GBM (Figure 1), an important event for immune complex deposition [104]. Alternative splicing may contribute to soluble PLA2R [105]. The cleavage of THSD7A at a site close to cell membrane may be in part attributable to soluble THSD7A [106].

## 5. Other MN-Associated Antibodies Targeting Podocyte Antigens

### 5.1. Exostosin 1 and Exostosin 2

By using laser capture-coupled mass spectrometry, exostosin 1 (EXT1) and EXT2 were identified in 5 of 15 PLA2R-negative MN and none in 7 PLAR2-positive MN cases in a pilot cohort [107]. In the discovery cohort of 224 PLA2R-negative MN cases and 102 controls (including 47 cases of PLAR2-positrive MN and other controls), mass spectrometry and immunohistochemistry (IHC) detected 21 and 26 (11.6%) EXT1/EXT2-positive cases, respectively, only in PLA2R-negative MN cases. Granular staining of EXT1 and EXT2 along the GBM and a typical staining pattern of MN was demonstrated. IgG1 was the most abundant anti-EXT1/EXT2. The deposition of component C3 along the capillary wall was observed [107]. Most patients (85%) with EXT1/EXT2-positive MN had other autoimmune diseases including lupus (8/26), revealing EXT1 and EXT2 as potential target antigens for SMN [107]. The association with pure class V lupus nephritis was confirmed in 8 out of 18 patients in a validation cohort [107]. EXT1/EXT2-positive SMN occurred predominantly in females (21/26 = 80.8%) [107]. While evidence favors EXT1 and EXT2 as potential target antigens for SMN, there are patients among these 26 EXT1/EXT2-positive MN cases with no evidence of existing autoimmune conditions [107].

EXT1 and EXT2 are members of the exostosin glycosyltransferase family; both function in the synthesis of the heparan sulfate (HS) backbone via chain elongation [108,109]. EXT1 (8q24.11) and EXT2 (11p11.2) form heterodimers, which enhances their activity and stability [110]. EXT1 and EXT2 form a complex in the endoplasmic reticulum (ER) transmembrane and loss of their functions causes hereditary multiple osteochondromas [111]. By functioning in HS synthesis, EXT1 and EXT2 affect multiple signaling events, including FGF, BMP, and Wnt signaling [111]. How these signaling pathways and the ER residence of EXT1 and EXT2 contribute to their impact on SMN are an interesting aspect of future research.

### 5.2. Neural Cell Adhesion Molecule 1 (NCAM-1)

Similar to the association of EXT1 and EXT2 with lupus nephritis [107], NCAM-1 was reported as an antigen for membranous lupus nephritis (MLN) and PMN [112]. In an effort to discover unknown MN antigens via mass spectrometry analysis of laser-capture micro-dissected MN glomeruli, NCAM-1 was identified in three cases of MLN [112]. Among 212 MLN and 102 PMN cases, 14 (6.6%) and 2 (2%) were associated with NCAM-1, respectively [112]. The co-localization of NCAM-1 with IgG along the GBM was observed. Both C1q and C3 were detected in immune deposits [112]. Among 13 cases with determined IgG subclasses, IgG1 was present in 11 cases with 6 cases being IgG1 dominant [112], showing that IgG4 is not the dominant anti-NCAM-1 antibody IgG. The IgG1 dominance for anti-NCAM-1 antibodies shares similarity with anti-EXT1/EXT2 antibodies [107], supporting the association of both NCAM-1 and EXT1/EXT2 with lupus nephritis. Nonetheless, aPLA2R-Ab was also reported in 5.3% of MLN cases and the presence of aPLA2R-Ab is associated with poor renal prognosis [113]. However, the clinical impact of anti-NCAM-1 antibodies needs further investigation. NCAM-1 is a neural cell adhesion protein with a high level of expression in the central nervous system, and its association with proliferative lupus nephritis was previously reported [114].

### 5.3. Neural Epidermal Growth Factor-like 1 (NELL-1)

Following the system described above, the same group recently identified NELL-1 as a target antigen candidate for PMN [115]. Among PLA2R-negative MN cases from a discovery cohort (*n* = 126) from the Mayo clinic and a validation population (*n* = 84) from France and Belgium, 16.2% (34/210) of NELL-1 positive cases were detected [115]. IgG1 was the most abundant Ig subclass produced in patients with NELL-1 MN. Granular staining of NELL-1 along the GBM was demonstrated; the co-staining of anti-NELL-1 with IgG supports anti-NELL-1 IgG as a component of IgG deposition in the subepithelial surrounding. The staining of component C3 occurred along the capillary wall [115]. Circulating anti-NELL-1 in five patients was demonstrated in serum [115]. NELL-1 is expressed at a high level in neural tissues including the brain and at a low level in non-neural tissues such as in the liver and kidney [116]. In the kidney, tubules express a higher level of NELL-1 and its expression is marginally detected in the glomeruli [115,117]. Nonetheless, the authors favored the possibility of NELL-1 shedding from the podocyte for the immune complex deposition along the GBM [115]. NELL-1 is a secreted protein [118]. With NELL-1 being expressed at higher levels in other tissues, together with its presence in the extracellular domain (secreted), and the circulating anti-NELL-1 antibody in MN patients, the possibility of the deposition of the anti-NELL-1-Ab-NELL-1 complex into to the subepithelial region cannot be excluded. Nonetheless, the evidence collectively supports NELL-1 as a MN antigen.

In a discovery cohort containing 126 patients with PLA2R-negative MN, 29 NELL-1 positive MN cases were not associated with autoimmunity and other system conditions, suggesting NELL-1 as a potential target antigen of PMN. However, this should be interpreted with caution; in the validation cohort (*n* = 84), 4 among 5 NELL-1 positive MN patients had cancer (lung cancer, metastatic pancreatic carcinoma, metastatic breast cancer, and urothelial carcinoma) [115].

The association of NELL-1 MN with cancer is in accordance with NELL-1 being a potent growth factor for the osteochondral lineage [118]. NELL-1 activates the mitogen-activated protein kinase (MAPK) pathway and Wnt/β-catenin signaling [119]; both are well-established oncogenic events [120,121]. NELL-1 contains a set of domains. The N-terminal thrombospondin-like module (TSPN) binds heparin [119,122], which facilitates NELL-1 association with integrin [123]. The upregulation of NELL-1 causes congenital cranial defects attributed to the premature fusion of sutures [124,125]; mice with transgenic overexpression of NELL-1 also develop craniosynostosis [126]. These observations are in line with the critical role of NELL-1 in osteogenesis and skeletal development [119]. EXT1 and EXT2 play important roles in HS synthesis and their loss leads to inherited skeleton defects [111,127]; EXT1/EXT2 and NELL-1 were identified in the same system and in PLA2R-negative MN. Additionally, autoantibodies to EXT1/EXT2 and NELL-1 in MN patients were IgG1-based. With these similarities, it is tempting to suggest at least some association between EXT1/EXT2 and NELL-1 in MN development.

With NELL-1 contributing to 5–10% of PMN cases along with PLA2R (70–80%) and THSD7A (1–5%), approximately 5–10% of PMN cases seem to involve antigens yet to be identified [128].

### 5.4. Contactin 1 (CNTN1)

Similar to NELL-1, Contactin 1 (CNTN1) is mainly expressed in neural tissues [129]. CNTN1 forms a complex with contactin-associated protein-1 (CASPR1) on the axonal membrane, which binds neurofascin (Nfasc) on the Schwann cell surface to form septate-like axoglial junctions in the paranodal region [130,131]. CNTN1 plays an essential role in maintaining the cohesion between the axon and the myelin sheath in the paranodal loops. Antibodies targeting CNTN1 and other two proteins (CASPR1 and Nfasc) have been reported in patients with chronic inflammatory demyelinating polyradiculoneuropathy (CIDP) [132,133,134,135], a heterogenous chronic autoimmune neuropathy characterized by autoimmunity-induced demyelination [136,137].

Evidence is accumulating for the association of CNTN1 with MN. Anti-CNTN1 (IgG4) was detected in 0.7% (*n* = 1500) [138], 0.8% (*n* = 342) [139], 6.5% (*n* = 46) [140], and 7.5% (*n* = 53) [141] of CIDP cases respectively. Among the 10 patients out of 1500 CIDP cases with IgG4 anti-CNTN1, 6 had concurrent MN; these patients were PLA2R-negative [138]. In a separate study, one of three anti-CNTN1 positive CIDP patients exhibited concurrent MN [139]. In addition to these 7 CNTN1-associated MN cases reported in 2020, 14 anti-CNTN1 positive CIDP patients with MN were accumulatively described from 1987–2018 [141,142,143,144,145,146,147,148,149,150,151,152], including 10 cases with neuropathy occurring before nephropathy and 4 cases with concurrent pathologies (please see literature review by Hashimoto et al.) [152]. CNTN1-associated MN has a male to female ratio of 10:4 (2.5:1) [152]. While the development of MN in these 21 CIDP patients with anti-CNTN1 Ab implies that kidney CNTN1 is a potential target, this has not been directly determined. This is important, as CNTN1 mRNA is only weakly expressed in the kidneys [153].

In a late case report of a 43-year-old male with CIDP who developed MN with nephrotic syndrome one year after being diagnosed with neuropathy, IgG4 anti-CNTN1 but not anti-PLA2R antibody was detected [154]. IgG deposition was confirmed by immunofluorescence staining. Clear podocyte expression of CNTN1 by IHC only occurred in the MN glomeruli but not in controls [154], providing direct evidence for podocyte CNTN1 upregulation as a potential target antigen for CIDP-associated MN. It is tempting to suggest podocyte CNTN1 as a major target antigen for MN occurring in CIDP patients. This concept is supported by the common observation that these patients are generally resistant to CIDP therapy such as intravenous immunoglobulin (IVIG) but sensitive to immunosuppressive treatments [138]. For this 43-year-old male with CIDP and MN, IVIG was not effective and immunosuppressive therapy improved both nephrotic syndrome and neuropathy [154].

MN development in the CIDP population appears specific for CNTN1. The CNTN1-Caspr1-Nfasc complex is required to form septate-like axoglial junctions in the paranodal loops [130,131]. While autoantibodies to this complex were produced, MN only occurred in 6 of 10 patients with IgG4 anti-CNTN1, despite 15 among 1500 CIDP patients producing anti-Nfasc antibodies [138]. This is more intriguing considering that podocytes express high levels of Nfasc in their major processes [155].

### 5.5. Semaphorin 3B, High Temperature Recombinant Protein A1 (HTRA1), and Protocadherin 7 (PCDH7)

Mass spectrometry analysis of laser dissected PLA2R-negative PMN biopsies identified semaphorin 3B in 3 cases among 70 biopsies in the Mayo clinical cohort [156]. Other cases (*n* = 8) were from French cohort #1 (2 out of 16 cases: 2/16), French cohort #2 (2/59), and an Italian cohort (4/43). IgG1 was the dominant subclass among 4 cases in which an IgG subclass was determined. Among the 11 cases of semaphorin 3B MN, 8 were pediatric patients; among the remaining 3 adult cases, the average age of onset was 36.3 years. The data suggests the occurrence of anti-semaphorin 3B IgG1 antibodies in pediatric and young MN patients. Complement C1q, C2, and 3 were detected in immune deposits [156]. Collectively, evidence supports semaphorin 3B as a new PMN antigen particularly for pediatric and young patients. Semaphorin 3B is a secreted protein; its expression in podocytes was reported [157]. Nonetheless, its functions in the kidneys remain unclear.

HTRA1 was recently identified as a target podocyte antigen in 3 PMN cases; anti-HTRA1 antibodies were predominantly IgG4 [158]. Among 85 PLA2R-negative PMN biopsies, PCDH7 was identified in 8 cases [159]. Granular deposits of PCDH7, IgG, and C3 along the GBM were observed; IgG typing on two cases revealed IgG4 in both [159]. A total of 12 PCDH7-associated MN cases have been accumulated with the mean age of 61 and sex ratio of 3:1 for males vs. females [160]. C1q and C3 deposition was absent. One of 12 patients had prostate cancer [160]. The evidence supports PCDH7 as a new PMN antigen. The involvement of HTRA1 and PCDH7 as PMN antigens requires additional investigations.

## 6. Other Aspects

Following the identification of PLA2R in 2009 and THSD7A in 2014, much has been done to demonstrate both PMN target antigens being present on the podocyte surface. The clinical potentials of PLA2R and THSD7A in terms of diagnosis, prognosis, and therapy responses have been demonstrated. However, it is important to note that high levels of aPLA2R-Ab can correlate with good prognoses; two patients with high levels of aPLA2R-Ab had spontaneous remission [161]; it was also observed that patients with aPLA2R-Ab+ MN had a higher remission rate than those with aPLA2R-Ab– MN following therapy [27]. These exceptions indicate complex mechanisms underlying PLA2R-caused PMN pathogenesis and progression, about which much less is known. The clinical applications of PLA2R and THSD7A can also be affected by the existence of PLA2R- and THSD7A-negative PMN cases.

This concern can be reduced with the latest evidence supporting NELL-1 as an antigen in PLA2R- and THSD7A-negative PMN cases. It appears that PMN cases with circulating IgG1 anti-NELL-1 constitute 5–10% of PMN [115,128]. Considering the distribution of currently identified MN antigens including PLA2R (70–80%), THSD7A (1–5%), NELL-1 (5–10%), NCAM-1, semaphorin 3B, HTRA1, PCDH7, and unknown antigens (5–10%) in PMN cases [128], we are getting much closer to mapping out the full spectrum of PMN target antigens.

However, the tasks ahead are certainly challenging. PMN constitutes 75% of MN cases, and can be classified as PLA2R-, THSD7A-, or possibly NELL-1-associated. Nonetheless, the existence of aPLA2R-Ab in 16% to 36% of SMN cases in some reports indicates a more complex scenario. Are there common causes for both PMN and SMN? This concept could be supported by the association of both IgG4 (PLA2R and THSD7A) and IgG1 (NELL-1) with PMN. Additionally, in a retrospective analysis of 58 MN biopsies, only PLA2R (*p* = 0.25) showed higher staining in PMN than SMN, but this was not the case for THSD7A (*n* = 6) and IgG4 (*n* = 34) staining [162]. Both PLA2R and THSD7A are well-established for their association with PMN. While the association of THSD7A with cancer was not observed in every study [12,52,54,55], reports are accumulating for its presence in cancer-associated SMN [163,164], which might be particularly relevant in colorectal and breast cancer-caused SMN [165]. In a study of malignancy-associated MN, 16 cancer cases with PLA2R-associated MN were reported [163].

Accumulative evidence supports complement activation as a potential mechanism of MN pathogenesis [95,97,166]. In this regard, attempts have been made to inhibit complement activation as a therapeutic option to treat MN. In view of the central position of C3 and C5 in complement activation via CP, LP, and AP, inhibitors to C3 and C5 have been developed. Eculizumab, a humanized monoclonal anti-C5 antibody (Alexion Pharmaceuticals, Boston, MA), was not effective in reducing proteinuria in MN patients [166]. Pegcetacoplan (APL-2; Apellis Pharmaceuticals, Waltham, MA) is a C3 inhibitor currently being examined in a Phase II clinical trial (NCT03453619) on glomerulopathies, including PMN. However, evidence suggests effects being non-satisfactory [166]. Other ongoing clinical trials in MN include inhibition-targeting APs and LPs. LNP023 (Novartis, Basel, Switzerland) is an inhibitor of complement factor B (CFB) and thus inhibits AP actions [167,168]; LNP023 is under a Phase II clinical trial in PMN patients (NCT04154787). Narsoplimab (OMS721; Omeros, Seattle, Washington, USA) is a monoclonal human IgG4 antibody inhibiting mannan-binding lectin-associated serine protease-2 (MASP-2) [169], a protease that cleaves C2 and C4 to initiate complement activation via an LP [166]. OMS721 is under a Phase II clinical trial in MN patients (NCT02682407). The inhibition of the initiation pathways regulating complement activation (APs and LPs) is supported by the predominance of IgG4 in PLA2R- and THSD7A-associated PMN patients and the involvement of both the AP and LP in PLA2R-associated PMN (see Section 4.3) [166]; these patients may benefit from LNP023 or OMS721 therapy. However, for MN cases involving IgG1, other approaches for targeting complement activation need to be explored.

## 7. Future Perspectives

Now is an exciting time to study PMN; the field is expeditiously expanding. Podocytes are likely subjected to primary injury during MN pathogenesis; it is interesting to see that iPSC (inducible pluripotent stem cell)-differentiated podocytes can repair the injury in a mouse model of MN [170]; whether this can be used alone or together with immunosuppressive therapy should be explored. While rituximab is emerging as the standard immunosuppressive therapy for PMN [21], the knowledge gained on PLA2R and THSD7A needs to be translated into the clinic for targeted therapies. To reach this status, the current understanding on MN requires substantial advancement.

Knowledge on NELL-1 contributions to MN or PMN will be rapidly emerging. Its relationship to other MN antigens can be studied. For instance, as a secreted protein or growth factor, its actions in osteogenesis require it to bind to a receptor, cantactin-associated protein-like 4 (CNTNAP4) [119]. It would be fascinating to examine the connection between NELL-1 and CNTN1 in MN pathogenesis.

While CNTN1 has been suspected to be an MN antigen in a sub-CIDP population for more than 30 years, its detection in podocytes in a patient with both CIDP and MN pathologies further supports this possibility. CNTN1 is anchored on the cell surface and contains multiple extracellular motifs (Figure 2D), which share structural similarities with PLA2R and THSD7A (Figure 2). CNTN1 modulates multiple cellular signaling pathways and facilitates tumorigenesis [171,172,173,174]. Whether CNTN1 is a novel MN antigen should be further explored.

MN-associated antigens in humans, PLA2R, NELL-1, and (possibly) CNTN1, are expressed at low levels in podocytes. It is thus tantalizing to propose that this likely neoantigen status in the podocyte is a contributing factor for MN. However, merely being a neoantigen is not sufficient. The transgenic expression of PLA2R in mouse podocytes does not by itself cause MN without the passive transfer of anti-PLA2R IgG [18], despite PLA2R being the most common PMN target antigen. Additionally, mice with a transgenic expression of human laminin α5 in the basement membrane only produce MN in transgenic fetuses produced from crosses between transgenic males and wild type females [175], which resulted from the transfer of maternal anti-human laminin α5 antibodies [175,176]. Another intriguing situation is that the immunization of DBA1 mice with a 14-mer peptide 23 of human COLIV α3 NCI protein produced comparable antibodies in all mice but only 50% of animals developed MN [177], suggesting the need for additional hits [178].

The lack of understanding on the other hits is a major hurdle for MN research and clinical care. The lung was suggested as being an initial site for producing PLA2R immunity, evident by PLA2R expression in the lung [105] and MN incidence association with air pollution [179]. Evidence now also supports the contribution of complement activation to MN pathogenesis [90,180]. Starting from the generation of the Heymann nephritis model in 1959 [9], the passive transfer of IgG remains the standard animal model in current MN research. The limitation of this model is evident; it cannot address the initial factors critical for MN development. A major risk factor of MN is aging; the rapid onset of passive MN models is not applicable when analyzing the time factor. New animal models may be needed to advance MN research to the next level.

## Figures and Tables

**Figure 1 biomolecules-11-00513-f001:**
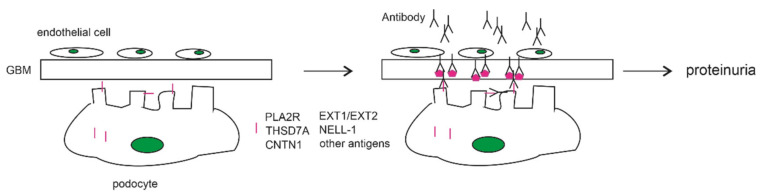
Factors promoting membranous nephropathy pathogenesis. Mechanisms regulate the production of autoantibodies to the podocyte antigens and the shedding of podocyte antigens into the GBM; this leads to the formation of immune complex in the subepithelial region and within the GBM (right panel), subsequently causing proteinuria.

**Figure 2 biomolecules-11-00513-f002:**
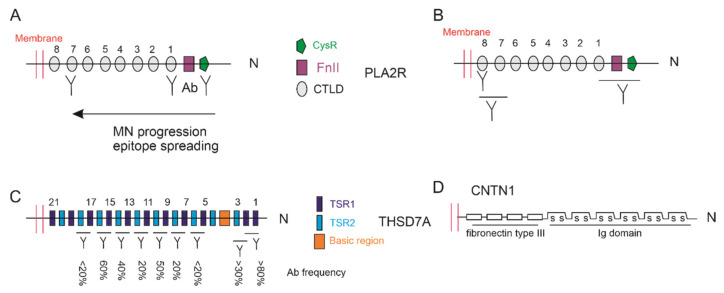
Epitope utilization for PLA2R and THSD7A in PMN patients. Both epitope spreading (**A**) and co-utilization of epitopes distant and adjacent to podocyte membrane (**B**) occurred in PLA2R-associated PMN patients. The epitope frequency for THSD7A (**C**) was derived from reference (79) based on 150 THSD7A-associated PMN cases. The 21 TSP-1 domains in TSD7A can be classified as group 1 thrombospondin repeat 1 (TSR 1) and group TSR 2 [57]. (**D**) CNTN1 is anchored on the cell surface.

**Table 1 biomolecules-11-00513-t001:** Diagnosis of primary membranous nephropathy (PMN) with serum anti-PLA2R antibodies.

PMN Cases	Control Cases ^1^	Sensitivity	Specificity	Refs
69	386	71%	100%	[39]
57	84	82.5%	75%	[40]
67	236	88.1%	96%	[41]
155	154	83.9%	99.4%	[42]
374	296	80.8%	98%	[43]

^1^ Control cases include secondary MN (SMN) cases, patients with non-MN renal disease, and healthy individuals.

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
