# Peer review of "Mechanisms of Primary Membranous Nephropathy"

_biomolecules, 2021, doi:10.3390/biom11040513_

Round 1

Reviewer 1 Report

This manuscript is a comprehensive review of the literature on new pathological mechanisms of primary membranous nephropathy in the last five years.
It is a very comprehensive review with more than 150 references published up to 2020.
I would like to ask the authors if they can add a few sentences about the recurrence of primary membranous nephropathy after renal transplantation, which is quite common and associated with an increased risk of allograft loss.
It is very unusual to write "Conclusions" with the references and describe a study (reference 136). In my opinion, the last paragraph of "Conclusions" is unnecessary.

Author Response

We appreciate the reviewer’s overall positive remarks; these comments contribute to the improvement of this manuscript. Here are our detailed revisions.

Reviewer comments – “I would like to ask the authors if they can add a few sentences about the recurrence of primary membranous nephropathy after renal transplantation, which is quite common and associated with an increased risk of allograft loss.”

Authors' response – We agree with the reviewer; the recurrence in patients with PLA2R- and THSD7A-associated MN provides functional support for the antibodies being causative to MN. Recurrent MN related to anti-PLA2R (lines 181-190, marked with red) and anti-THSD7A antibody (lines 246-248, marked with red) are described.

Reviewer comments – “It is very unusual to write "Conclusions" with the references and describe a study (reference 136). In my opinion, the last paragraph of "Conclusions" is unnecessary”.

Authors' response – We thank the reviewer for this insightful structural comment. The intention of section 6 was to outline the complex nature, exceptions, and other aspects related the concepts developed in previous sections. For instance, anti-PLA2R antibody is not always associated with unfavorable outcomes of PMN; and the potential connections between PMN and secondary MN (SMN). We realize these contents may not necessarily count as “Conclusions”. In this regard, we have changed the section title from “Conclusions” to “Other aspects”. As well, the first paragraph has been simplified. A paragraph on targeting complement activation for MN therapy has been added in response to Reviewer #3 comments. The revised section is better articulated with its content fitting the title.

Reviewer 2 Report

Overall a comprehensive review of an interesting topic. 
The aim of the paper seems to be to review the mechanisms at the basis of primary Membranous Nephropathy.
The article is very detailed and descriptive and I enjoyed reading it and learning from it. Overall, good review of prior research enlighting 
the role of the whole panel of antibodies used as biomarker of primary MN and as indicator of treatment response and disease activity. 

The authors review prior research, outlining data that support the premise of their paper and data that contradicts the premise.
I support the publication of this paper in the current version. 

Author Response

We appreciate the reviewer’s encouraging comments and are delighted that Reviewer #2 accepted the manuscript at the current form.

Reviewer 3 Report

This article is focused on the role of several autoantibodies that were recently identified in membranous nephropathy (MN). These antibodies are not only useful as biomarkers for diagnosis and monitoring disease activity, but they seem to be associated with the molecular pathogenesis of MN. I think is article is well written and incorporates recent and appropriate references. Basically, I have no substantial criticism except for one brief suggestion that might add some significance to the article.

Although the actual molecular events linking the autoantibodies with the development of proteinuria remains largely unknown, complement activation has been postulated as one potential mechanism of the pathogenesis of MN (1)(2). Several complement inhibitors are currently evaluated in clinical trials of MN as well (3)(4). I recommend the authors to include a paragraph discussing the possible role of complement system in the antibody-mediated glomerular injury in MN.

(1) Seikrit C et al. N Engl J Med. 2018, 379: 2479-81.

(2) Zhang MF et al. BMC Nephrol. 2019, 20: 313.

(3) https://clinicaltrials.gov/ct2/show/NCT02682407

(4) https://clinicaltrials.gov/ct2/show/NCT03453619

Author Response

We thank the reviewer for overall positive tone. Here are our detailed revisions.

Reviewer comments – “Although the actual molecular events linking the autoantibodies with the development of proteinuria remains largely unknown, complement activation has been postulated as one potential mechanism of the pathogenesis of MN (1)(2). Several complement inhibitors are currently evaluated in clinical trials of MN as well (3)(4). I recommend the authors to include a paragraph discussing the possible role of complement system in the antibody-mediated glomerular injury in MN.

(1) Seikrit C et al. N Engl J Med. 2018, 379: 2479-81.

(2) Zhang MF et al. BMC Nephrol. 2019, 20: 313.

(3) https://clinicaltrials.gov/ct2/show/NCT02682407

(4) https://clinicaltrials.gov/ct2/show/NCT03453619”

Authors' response – We appreciate these insightful comments; the references enclosed are very helpful. The knowledge of refs (1) and (2) is included in section 4.3 (lines 380-383, ref 94; lines 385-387, ref 96). A new paragraph has been organized to discuss the therapeutic potential of complement activation in MN therapy and all four references are discussed (lines 599-619).

Reviewer 4 Report

My comments are describe in the enclosed file

Author Response

We appreciate the reviewer’s careful evaluation. Our detail responses are outlined below.

Reviewer comment #1 – “Introduction is too long and not well structured. In fact, it should concentrate on basic facts on membranous nephropathy (pathology, clinical signs, outcome and treatment) and distinguish between primary and secondary membranous nephropathy, discovery of anti-podocyte antibodies should be already the next independent chapter”.

Authors' response – We see the reviewer’s position. The focus of this review is to discuss recent developments (approximately in the past 5 years) related to mechanisms underpinning primary membranous nephropathy. In this regard, we believe the contents of “Introduction” provide a proper framework for the sections to follow. Brief discussions on the clinical features, PMN and SMN, and fundamental works leading to the discoveries of MN antigens in humans are included (lines 29-42). We trust the Reviewer #4 will see our position.

Reviewer comment #2 – “The overview of anti-podocyte antibodies should include recent discoveries of anti-NCAM1, anti-semaphorin 3B and anti-HTRA1 antibodies. The role of these autoantibodies in specific subtypes of MN should be described (pediatric, segmental, lupus MN), this should be added to the chapter „Other MN-associated podocyte antigens“ (in fact, the authors should better speak about „Other anti-podocyte antibodies”.

Authors' response – We appreciate these insightful remarks and the literature information provided. As the reviewer suggested, the title of section 5 has been revised from “Other MN-associated podocyte antigens” to “Other MN-associated antibodies targeting podocyte antigens”. The recent discoveries on NCAM-1, semaphorin 3B, HTRA1, and protocadherin 7 (PCDH7) as new MN antigens have been organized into 2 new subsections: 5.2 (lines 445-461, marked with red) and 5.5 (lines 545-566, marked with red).

Reviewer comment #3 – “Data on antibodies to contactin 1 are not well described – are there any non-Chinese papers on this subject, or do the authors believe that these antibodies may be specific for Han Chinese? Are there any genetic studies in pts with anti-contactin-1 Ab in membranous nephropathy?”

Reviewer comment #4 – “Generally Chinese and non-Chinese references should be more balanced and it should be clear which data were confirmed in different ethnicities”

Authors' response – Frankly, we do not know which scientific aspects of our manuscripts were commented on here. Which citations and how many citations in the CNTN1 section (5.4) were “Chinese papers”? The impartial selection process for articles to be reviewed was outlined (lines 74-77); not sure where and how Reviewer #4 came to the conclusion of biased selection towards “Chinese references”.

Reviewer comment #5 – “The role of complement activating and non-complement activating anti-podocyte antibodies should be delineated and the role of complement in MN should be described”

Authors' response – We thank the reviewer for the comments. A new section on complement activation in PLA2R- and THSD7A-associated MN has been added, section 4.3 (lines 363-393, marked with red).

Reviewer comment #6 – “Paragraph on the role of anti-PLA2R titre in the response to treatment should be possibly the part of independent chapter as it is not directly related to the pathogenesis of MN (page 3). Description of studies related to the anti-PLA2R Ab related response to treatment are too detailed and should shortened and summed up. Moreover, it should be mentioned that anti-PLA2R titres are also related to the renal outcome and spontaneous remission”

Authors' response – The associations of anti-PLA2R antibodies with clinical features and outcomes are highlighted (lines 120-125, marked with red; lines 130-142, marked with red).

Reviewer comment #7 – “It should be mentioned that epitope spreading of PLA2R (page 8) is related to the outcome and response to treatment”

Authors' response – We thank the reviewer for the comments. The concept has been discussed (lines 326-330, marked with red).

Reviewer comment #8 – “The final chapter „Future perspectives“ should be more balanced – half of the chapter is related to anti-contactin 1 antibodies. Here the authors should discuss the role of anti-podocyte antibodies in the non-invasive diagnosis, monitoring of activity, predicting therapeutic response, personalizing treatment, outcome, etc.”

Authors' response – While we do not share the Reviewer’s assessment of “half of” the contents being contactin 1-related, we made efforts to further reduce this content in section 7. The 4th paragraph has been simplified by removing this content (lines 641-653, marked with red).

Reviewer minor comment #1 – “English needs editing by native speaker, some terms are not appropriate (e.g. page 3 enrichment of anti-PLA2R Ab in primary membranous nephropathy)”

Authors' response – We thank the reviewer for pointing out the non-precise statement. This has been revised (lines 109-110, marked with red). The manuscript has been edited by a colleague of native speaker.

Reviewer minor comment #2 – “References – it is not necessary to mention (repeatedly) that AJKD is the official journal of NKF and NDT is an official journal of ERA-EDTA”

Authors' response – This redundance was produced by the citation program. We will fix this at galley proof stage later.

Round 2

Reviewer 4 Report

The authors adequately responded to almost all my comments with the exception of the question related to the anti-contactin antibodies. My question was if there are any non-Chinese data on these autoantibodies, so if this observation may be generalized, or if it seems to be specific for Han Chinese